# Time Window of Perturbation-Induced Response Triggered by Ankle Motion and Body Sway above the Ankle

**DOI:** 10.3390/brainsci10040230

**Published:** 2020-04-11

**Authors:** Koichi Hiraoka, Toru Kuramitsu, Naoki Nii, Miyuki Osumi, Nana Tanaka

**Affiliations:** College of Health and Human Sciences, Osaka Prefecture University, Habikino City 583-8555, Japan; kenken312343@yahoo.co.jp (T.K.); kyakya312343@yahoo.co.jp (N.N.); hayahaya312343@yahoo.co.jp (M.O.); nozanoza312343@yahoo.co.jp (N.T.)

**Keywords:** perturbation, postural control, soleus, gastrocnemius, electromyography, somatic sensation

## Abstract

We determined the precise time windows of the electromyographic (EMG) response components triggered by ankle motion and by body sway above the ankle. A support surface under the feet of healthy young adult participants in the quiet stance was moved in translation. The EMG response component triggered by body displacement above the ankle began at 95–100 ms and ended 145–155 ms after the onset of the support surface translation. The EMG response triggered by ankle dorsiflexion began at 35–50 ms and ended 110–115 ms after the onset of the translation in the soleus muscle, indicating that the response component began at a time similar to the short-latency response. In contrast, the response component in the gastrocnemius muscle began noticeably after that. The EMG response triggered by ankle dorsiflexion began at 75–85 ms and ended 125–135 ms after the onset of the translation in the gastrocnemius muscle. Our findings indicate that the threshold of the early response component to the somatic sensation of the ankle motion in the soleus muscle is lower than that in the gastrocnemius muscle. The response component triggered by the ankle motion continued long after the end of ankle dorsiflexion, indicating that the early component is mediated not only by the monosynaptic stretch reflex pathway but also by the polysynaptic pathway.

## 1. Introduction

Several studies have investigated the electromyographic (EMG) response components triggered by somatic sensations from particular body parts [1,2,3]. Those studies reported that the EMG response 40–100 ms after the onset of the postural perturbation was triggered by the ankle motion, but that the response from 100–220 ms after the onset was triggered by the body sway above the ankle. There were weaknesses in these previous studies. One weakness was that the precise time windows of the EMG response components triggered by the ankle motion and triggered by the body sway above the ankle were not elucidated. Another weakness was that slight ankle motion was still present in the nulled ankle input task, in which the ankle motion to the postural perturbation was minimal due to the simultaneous provision of backward translation and the toe-down tilt of the support surface. The postural perturbation was induced by a combination of horizontal translation and rotation of the support surface in the enhanced and nulled ankle input tasks. This caused the interpretation of the findings to be difficult, as these two support surface movements induced different postural responses [4,5]. Accordingly, the time window of the EMG response component triggered by the somatic sensation derived from a particular body part is still a matter of investigation.

In the present study, we examined the precise time windows of the EMG response components in the ankle extensors triggered by the ankle motion and triggered by the pelvis sway above the ankle during the horizontal translation of the movable platform. Through estimating these precise time windows, the relationship between the EMG response components triggered by the particular source of the somatic sensation and the short- and middle-latency EMG responses, distinguished by the onset latency, were examined.

In order to elucidate this, the amplitude of the EMG response was measured for both unilateral and bilateral (BL) translation of the platform, as per the method of Dietz et al. (1989) [6]. The tasks used in the present study are shown in Figure 1A. The unilateral tasks consisted of the unilateral left (ULL) task, translating only the support surface under the left foot, and the unilateral right (ULR) task, translating only the support surface under the right foot. In those tasks, one foot was placed over the movable platform, inducing a dorsiflexion movement at the ankle (unilateral moving ankle; UM ankle), with the contralateral foot on a non-movable platform, in which the ankle was not moved (unilateral stationary ankle; US ankle). In the BL task, both feet were placed over the movable platform, inducing a dorsiflexion movement of both ankles (B ankles).

The amplitude of the anterior–posterior pelvis sway in the BL task was presumed to be significantly greater than in the ULL and ULR tasks because the pelvis sway induced by the translation of the support surface under both feet is thought to be greater than that induced by the translation of the support surface under one foot. The velocity of the dorsiflexion in the UM ankle was presumed to be greater than in the US ankle. The velocity in the US ankle is almost zero, as the support surface under the US ankle does not move while the support surface under the UM ankle moves. In addition, as the velocity and amplitude of the platform translation are the same for the unilateral and BL tasks, the velocity of ankle motion-induced must be equivalent between the B and UM ankles.

Under these assumptions, the EMG amplitude averaged in each 5-ms epoch was statistically compared between the B and UM ankles for the same side of the leg to determine the time window of the EMG response component triggered by the body sway above the ankle (see Figure 1). The time epochs, in which the EMG amplitude is significantly greater for the B ankles than for the UM ankle, reflects the time window of the EMG response component triggered by the body sway above the ankle. This is because the pelvis sway, representing the body sway above the ankle, is greater in the B ankles than in the UM ankle; however, the velocity of the ankle dorsiflexion is equivalent between the B and UM ankles according to our assumption.

The amplitude of the EMG response was statistically compared between the UM and US for the same side of the ankle in each 5-ms epoch. According to our assumption, the ankle dorsiflexion must be present in the UM ankle but absent in the US ankle, and the pelvis sway is equivalent between the UM and US ankles. Based on this view, we expected that time epochs of the EMG response, in which the EMG amplitude in the UM ankle was significantly greater than that in the US ankle, represented the time window of the EMG response component triggered by the ankle motion.

One issue investigated through this study design is whether the response component triggered by the ankle motion is mediated only by the monosynaptic stretch reflex pathway or mediated by both the monosynaptic and polysynaptic pathways. The short-latency response is mediated by group Ia muscle afferents [7], being induced at a latency of 35–45 ms [7,8,9,10]. The latency of the T-wave, mediated by the stretch reflex pathway, is also about 35 ms [11]. Therefore, the earliest component of the EMG response in the soleus muscle is likely mediated by the group Ia afferents contributing to the monosynaptic stretch reflex. Based on this, if the EMG response component triggered by the ankle motion terminates within 30–40 ms after the end of the ankle motion, then the response component triggered by the ankle motion is considered to be mediated by the monosynaptic stretch reflex pathway contributing to the short-latency response [8].

On the other hand, there is a possible view that the early response component triggered by the ankle motion is mediated not only by the monosynaptic reflex pathway but also by the polysynaptic pathway. Postural perturbation induces the deviation of the center of mass, which causes a cutaneous sensation of the foot sole. The cutaneous sensation is mediated by the polysynaptic pathway causing a longer conduction time [12]. Thus, such a polysynaptic pathway mediating the cutaneous sensation may also mediate the early response component mediated by the ankle motion. If this response component continues later than 40 ms after the offset of the ankle motion, then the response component is not only mediated by the monosynaptic reflex pathway, but also by the polysynaptic pathway (hypothesis 1).

The contraction velocity of the gastrocnemius muscle is faster than that of the soleus muscle during walking [13]. The motor unit property and motor unit activity are different between the gastrocnemius and soleus muscles [14,15]. The vestibular contribution to the motor unit activity in the gastrocnemius muscle is greater than in the soleus muscle during postural tasks [16]. The gastrocnemius muscle contracts intermittently during postural tasks, while the soleus muscle contracts continuously [17]. Accordingly, there are physiological differences between the gastrocnemius and soleus muscles.

The probability of the presence and amplitude of the short-latency response in the soleus muscle are greater than those in the gastrocnemius muscle [10]. The soleus muscle receives greater feedback from the muscle spindles compared to the gastrocnemius muscle [18]. Moreover, the number of muscle spindles in the soleus muscle is greater than in the gastrocnemius muscle [19]. According to those previous findings, the afferent feedback from the muscle spindle in the soleus muscle is greater than that in the gastrocnemius muscle. Therefore, the threshold of the response to the somatic sensation caused by the ankle motion in the soleus muscle is likely lower than in the gastrocnemius muscle. Based on this view, we hypothesized that the onset of the EMG response component triggered by the ankle motion in the soleus muscle is earlier than that in the gastrocnemius muscle due to the lower threshold of the response to the somatic sensation of the ankle motion (hypothesis 2).

## 2. Materials and Methods

### 2.1. Participants

Our participants were 15 healthy young adults (9 males and 6 females), who were 19.9 ± 2.1 years old. The footedness score assessed by Waterloo Footedness Questionnaire-Revised was 8.6 ± 1.4 [20,21]. Fourteen participants were right-footed, and one was left-footed. They had no history of orthopedic or neurological diseases. The study protocol was approved by the Graduate School of Comprehensive Rehabilitation, Osaka Prefecture University Committee on Research Ethics (Approve number; 2018-107), and all participants provided informed consent.

### 2.2. Measurements

An accelerometer (AS-10TB, Kyowa Dengyo, Chofu city, Tokyo, Japan) was placed over the skin at the midpoint of a line connecting the posterior superior iliac spine, bilaterally, to measure the anterior–posterior pelvis sway. Measurements of the pelvis sway or the lower lumbar spine in the stance position and during walking have previously been measured using an accelerometer [22,23,24]. The signals from the accelerometer were amplified (CDV-700A, Kyowa Dengyo, Chofu city, Tokyo, Japan) and high-pass filtered (1 kHz cutoff). Surface electrodes for the EMG recording (10 mm diameter) were placed over the belly of the medial gastrocnemius and soleus muscles in the direction of the muscle fibers, 2 cm apart, bilaterally. The electrodes measuring the EMG signals from the soleus muscle were placed slightly medial to the medial border of the gastrocnemius muscle. The EMG signals were amplified and band-pass filtered (15 Hz and 1 kHz cutoffs; MEG-2100, Nihon Kohden, Tokyo, Japan). An electrogoniometer (PH-412B, DKH, Nerima district, Tokyo, Japan) was attached over each ankle to measure the ankle motion on the sagittal plane. The analog signals from the surface electrodes, the accelerometer and the electrogoniometer, in the time window between 100 ms before and 1000 ms after the onset of the platform translation, were digitized at a sampling rate of 1 kHz (PowerLab/8sp, ADInstruments, Colorado Springs, CO, USA) and stored for offline analysis.

### 2.3. Procedure

Participants were informed that the support surface of the platform moved in translation backward. This instruction was to maintain the equal influence of the direction prediction on the response across the trials. In addition, they were asked to maintain an upright standing position against the platform translation. We did not inform the participants as to the velocity of the platform translation. The participants closed their eyes to exclude the influence of the visual input on the response and maintained a quiet standing position, with their feet 20 cm apart. An experimenter triggered the backward platform translation when the body sway in the participant was at a minimum. The direction of the platform translation was always backward to induce the response in the soleus and gastrocnemius muscles [25]. The participants stepped off the platform after the platform translation, stepped back on the platform about 10 s later, and closed their eyes with a quiet stance for the next trial.

For the BL task, both feet were placed on the movable platform (Uchida Denshi, Hachioji city, Tokyo, Japan). Two unilateral tasks were performed, one with the left foot on the movable platform and the right foot on the non-movable platform (ULL), and the other with the right foot on the movable platform and the left foot on the non-movable platform (ULR) (Figure 1A). The heights of the movable and non-movable platforms were the same. To induce a forward perturbation to stance, the movable platform translated backward under two velocities, slow and fast. In the slow condition, the platform translated backward at a peak velocity of 40 cm/s, over a distance of 15 mm, with a duration of 75 ms, while in the fast condition, the peak velocity of translation was 91 cm/s, over a distance of 35 mm, with a duration of 80 ms.

We ensured that the duration of translation was roughly matched between the slow and fast translation, as the duration of the platform translation affects the EMG response [26]. The time to peak velocity was 34 ms in the fast and 41 ms in the slow. Two velocity conditions were used because the ankle extensor response is dependent on the velocity of the horizontal translation of the support surface [26]. Eleven trials were completed, consecutively, for each of the six task conditions (two velocities X three tasks). The order of presentation of the task conditions was randomized across participants.

### 2.4. Data Analysis

The EMG traces were adjusted for the baseline offset and rectified. The EMG traces of the last ten trials in each set were time-locked to the trigger of the platform translation and were averaged. The EMG amplitude was averaged across the time window over the 100-ms epoch before the trigger of platform translation as a measure of the baseline EMG amplitude (Figure 1B). There was a set delay of 10 ms between the trigger and the platform translation onset according to measurements by an accelerometer placed over the platform in a preliminary experiment. Thus, the EMG amplitude was averaged over each 5-ms epoch, not from the trigger of the platform translation but from the platform translation onset (time 0) to 200 ms after the platform translation onset.

The onset and peak latency of the ankle dorsiflexion were visually estimated. The averaged velocity of the ankle dorsiflexion was estimated by the slope of the regression line (degrees/s) for the ankle motion trace in the time window between the onset and peak dorsiflexion. The onset and peak latency of the ankle dorsiflexion for the UM ankle was used as the time window for the velocity of the US ankle. The accelerometer signals were integrated twice to estimate the anterior–posterior pelvis sway [22]. The onset and peak latency and the amplitude of the pelvis sway were quantified, where the pelvis sway represented the body sway above the ankle.

A repeated-measures two-way analysis of variance (ANOVA) was conducted to test the effects of the platform velocity and task (two levels of velocity—slow and fast; and three levels of tasks—BL, ULL, and ULR) on the sway amplitude and the onset and peak latencies of the pelvis. For the velocity and the onset and the peak latencies of ankle dorsiflexion, two levels of platform velocity (slow and fast) and three levels of the ankle effect (B, UM, and US) were tested by ANOVA. If the ANOVA revealed a significant interaction between the main effects, then a test of the simple main effect followed by the multiple comparison test (Ryan’s method) was conducted.

A repeated-measures two-way ANOVA was conducted on the EMG amplitude. The ANOVA was conducted on each side of the muscle, as the EMG amplitude is statistically comparable only for the data on the same side of the same muscle. One main effect was the time effect, which included 41 levels of the time epoch (baseline EMG, 0–5 ms, 5–10 ms, .... 195–200 ms after the platform translation onset) and another main effect was three levels of ankle effect (B, UM, and US) (Figure 1B). When a significant interaction between the two main effects was identified, the test of the simple main effect was conducted for the ankle effect in each level of time epoch (41 levels) and for the time effect in each level of the ankle effect (B, UM, and US).

The significance of the time effect, particularly between the baseline EMG amplitude and the EMG amplitude after the platform translation onset revealed by the test of the simple main effect, was used to determine the time epochs in which the EMG response was present. Then, the period between the beginning and the end of the sequential time epochs, in which the significant effect was present, was considered to be the time window in which the EMG response was present. The beginning of the time window was considered to be the onset of the EMG response and the end was considered to be the offset of the EMG response.

If the test of the simple main effect revealed a significant ankle effect in some levels of the time epoch, then a multiple comparison test (Ryan’s method) was conducted for those time epochs to test the difference in EMG amplitude for each pair of the ankle conditions (B vs. UM, B vs. US, and UM vs. US).

The significance level was set at 0.05. Excel-Toukei 2010 ver. 1.13 (Social Survey Research Information, Tokyo, Japan) and ANOVA4 on the web (© Kiriki Kenshi 2002; www.hju.ac.jp/~kiriki/anova4) were used for the statistical analysis. The mean and standard error of the mean were used to express the data.

## 3. Results

### 3.1. Pelvis Sway

The traces of the pelvis sway from one participant are shown in Figure 2A, and the average sway onset latency and the latency of the peak sway across the participants are shown in Figure 3. During the backward platform translation, the pelvis was initially displaced in a backward direction, with this movement reversing about 130 ms after the onset of platform translation.

The average onset latency of the pelvis sway was 31 ms in the slow and 28 ms in the fast. The onset latency was significantly affected by the velocity of platform translation (F(1, 14) = 4.862, *p* = 0.045) but was not significantly affected by the task (F(2, 28) = 0.316, *p* = 0.732) without a significant interaction (F(2, 28) = 0.547, *p* = 0.585). The average latency of the peak pelvis sway was 60 ms in the slow and 57 ms in the fast. The latency was significantly affected by the velocity of platform translation (F(1, 14) = 25.402, *p* < 0.001), but was not significantly affected by the task (F(2, 28) = 0.495, *p* = 0.615) without a significant interaction (F(2, 28) = 0.426, *p* = 0.658).

The average amplitude of the pelvic sway across the participants is shown in Figure 4A. The amplitudes were 2.4 ± 0.2 cm in the BL, 1.2 ± 0.1 cm in the ULL, and 1.2 ± 0.1 cm in the ULR in the slow condition. In the fast condition, the amplitudes were 5.8 ± 0.6 cm in the BL, 3.0 ± 0.3 cm in the ULL, and 3.1 ± 0.3 cm in the ULR. The amplitude of pelvis sway was significantly influenced by the velocity of the platform translation and the task (see Table 1). There was a significant interaction between these two main effects. The test of the simple main effect revealed that the amplitude of pelvis sway was significantly greater for the fast than slow platform translation across the three tasks (BL, ULL, and ULR; *p* < 0.001). The main effect of the task was significant under both velocities of platform translation (*p* < 0.001). The multiple comparison test revealed that the amplitude of the pelvis sway was significantly greater for the BL than for the ULL and ULR tasks at both velocities of platform translation (*p* < 0.001), with no significant difference between the ULL and ULR tasks.

### 3.2. Ankle Motion

The traces of the ankle motion from one participant are shown in Figure 2B. In the moving ankle (B and UM), the dorsiflexion occurred during the backward translation of the platform, with no motion of the US ankle. The average onset and peak latency of the ankle dorsiflexion across the participants are shown in Figure 3. The average latency of the peak dorsiflexion ranged from 56 to 61 ms after the platform translation.

The average onset latency of the left ankle dorsiflexion was 18 ms in the slow translation and 15 ms in the fast. The onset latency of the left ankle dorsiflexion was significantly affected by the velocity of the platform translation (F(1, 14) = 18.839, *p* < 0.001) but was not significantly affected by the ankle effect (F(1, 14) = 1.497, *p* = 0.241) without a significant interaction (F(1, 14) = 0.961, *p* = 0.344). The average peak latency of the left ankle dorsiflexion was 60 ms. The peak latency was not significantly affected by the velocity of the platform translation (F(1, 14) = 0.020, *p* = 0.890), but was significantly affected by the ankle effect (F(1, 14) = 18.533, *p* < 0.001) without a significant interaction (F(1, 14) = 0.407, *p* = 0.534).

The average velocity of the left ankle dorsiflexion across the participants is shown in Figure 4B. The mean velocity was 57.1 ± 5.6 degrees/s in the B ankles, 60.7 ± 5.9 degrees/s in the UM ankle, and 0.8 ± 0.5 degrees/s in the US ankle for the slow translation. The mean velocity was 122.6 ± 10.6 degrees/s in the B ankles, 137.2 ± 11.7 degrees/s in the UM ankle, and 1.6 ± 0.8 degrees/s in the US ankle for the fast translation. The velocity of the left ankle dorsiflexion was influenced by a significant primary effect of the velocity of the platform translation and an ankle effect (B, UM, and US), with a significant interaction between these main effects (Table 2).

The test of the simple main effect revealed a significantly higher velocity of the ankle dorsiflexion for the fast platform compared with the slow platform translation for the B and UM ankles (*p* < 0.001). The main ankle effect was present for both velocities of the platform translation (*p* < 0.001). The multiple comparison test confirmed that the velocity of the dorsiflexion for the B and UM ankles was significantly greater than for the US ankle for both the slow and fast platform translations (*p* < 0.001). There was no significant difference in the velocity of the dorsiflexion between the B and UM ankles at either the slow (*p* = 0.637) or fast (*p* = 0.057) platform translations.

The onset latency of the right ankle dorsiflexion was 17 ms in the slow translation and 14 ms in the fast translation. The onset latency of the right ankle dorsiflexion was significantly affected by the velocity of the platform translation (F(1, 14) = 12.359, *p* = 0.003) and by the ankle effect (F(1, 14) = 11.156, *p* = 0.004) without a significant interaction (F(1, 14) = 1.471, *p* = 0.245). The average peak latency of the right ankle dorsiflexion was 57 ms. The peak latency of the right ankle dorsiflexion was not significantly affected by the velocity of platform translation (F(1, 14) = 1.725, *p* = 0.210), but was significantly affected by the ankle effect (F(1, 14) = 11.355, *p* = 0.005) without a significant interaction (F(1, 14) =1.694, *p* = 0.214).

The average velocity of right ankle dorsiflexion across the participants is shown in Figure 4C. The mean velocity was 55.5 ± 3.8 degrees/s in the B ankles, 56.7 ± 3.6 degrees/s in the left UM ankle, and 0.4 ± 0.3 degrees/s in the right UM ankle for the slow translation. The mean velocity was 105.8 ± 6.7 degrees/s in the B ankles, 108.7 ± 10.2 degrees/s in the UM ankle, and 1.3 ± 0.6 degrees/s in the US ankle for the fast translation. The velocity of the right ankle dorsiflexion was also influenced by the main effect of the velocity of the platform translation and ankle effect, with a significant interaction between these main effects (Table 2). The test of the simple main effect revealed a significantly greater velocity of dorsiflexion for the fast translation compared with the slow platform translation for the B and UM ankles (*p* < 0.001). The main effect of the ankle was present for both the slow and fast platform translations (*p* < 0.001). For the left ankle, the multiple comparison test confirmed a significantly greater velocity of dorsiflexion for the B and UM ankles than for the US ankle at both velocities of platform translation (*p* < 0.001). However, there was no significant difference between the B and UM ankles at either the slow (*p* = 0.870) or fast (*p* = 0.689) velocity of platform translation.

### 3.3. Average EMG Traces

The EMG traces, averaged across all participants, are shown in Figure 5. The largest EMG response was observed in the B ankles, with the smallest response in the US ankle. The EMG responses tended to be larger for the fast compared to the slow platform translations. Two peaks of the EMG response were observed in the soleus muscle for the slow platform translation, with no occurrence of the two peaks in the gastrocnemius muscle. The amplitude of the early component of the EMG response was similar for the B and UM ankles, and of much lower amplitude in the US ankle. For the late response, the amplitude was higher for the B than UM or US ankle, with no difference between the UM and US.

### 3.4. Time Epochs of EMG Response

The ANOVA results for the EMG amplitude averaged in each time epoch are summarized in Table 3. There was a significant main effect of the ankle (B, UM, and US) and the time to the platform translation onset with a significant ankle X time interaction on the EMG amplitude, both in the gastrocnemius and soleus muscles. The test of the simple main effect revealed a significant main effect of time for all tasks (*p* < 0.05). The multiple comparison test revealed that the EMG amplitude in some time epochs after the onset of the platform translation was significantly higher than the baseline EMG amplitude (*p* < 0.05). Those time epochs indicated the periods in which the EMG response was present.

The specific time epochs where the EMG amplitude after the platform translation onset was significantly higher than the baseline EMG are summarized in Figure 6. For the gastrocnemius muscle, the EMG response, in which the amplitude was significantly higher than the baseline, began at 80–90 ms after the onset of the platform translation for the B and UM ankle, and at 95–100 ms after that for the US ankle (*p* < 0.05). The EMG response ended 155–160 ms after the onset of platform translation for the B ankles, and 135–150 ms for the UM and US ankles (*p* < 0.05). For the soleus muscle, the EMG response began 40–55 ms after the onset of platform translation for the B and UM ankles, and from 105 ms after that for the US ankle (*p* < 0.05). The EMG response ended 145–155 ms after the onset of platform translation for the B ankles, and 125–140 ms after that for the UM and US ankles (*p* < 0.05). There was no period in which the EMG amplitude after the onset of platform translation was significantly lower than the baseline EMG.

### 3.5. EMG Response Triggered by Ankle Motion

The test of the simple main effect indicated significant ankle effects (B, UM, and US). Based on this result, a multiple comparison test was conducted (Figure 7). The EMG amplitude was significantly higher in the UM than in the US ankle (ankle factor; the EMG response component triggered by the ankle motion) over the time period beginning at 75–85 ms after the onset of platform perturbation for the gastrocnemius muscle, and 35–50 ms for the soleus muscle. The time window over which the EMG amplitude was significantly higher for the UM than for the US ankle ended at 125–135 ms after the onset of platform translation for the gastrocnemius muscle and 110–115 ms after that for the soleus muscle. In one exception, the EMG amplitude was significantly higher in the UM than in the US ankle over the period 130–135 ms after the onset of the platform translation for the right soleus muscle with the slow platform translation.

### 3.6. EMG Response Triggered by Body Sway

The time window of the EMG response component triggered by the body sway above the ankle (body sway factor), represented by a significantly higher EMG amplitude in the B than the UM ankle, is shown in Figure 7. The time window over which the EMG amplitude was significantly higher in the B than the UM ankle began from 90–100 ms after the onset of platform translation for both the gastrocnemius and soleus muscles, with the exception of the 55–60 ms time period, in which the EMG amplitude was significantly higher in the B than UM ankle for the soleus muscle with the fast platform translation. The time window in which the EMG amplitude of the UM was significantly higher than the US ankle ended 155 ms after the onset of platform translation for the gastrocnemius muscle and 145–155 ms after for the soleus muscle. The EMG at baseline was not significantly different between the ankle conditions.

## 4. Discussion

### 4.1. Methodological Consideration

In the present study, we determined the time window of the EMG response component in the gastrocnemius and soleus muscles triggered by the ankle motion and the response component triggered by the body sway above the ankle during a backward translation of the platform. Previous studies determined the latency of the EMG response using visual inspection [5,10], deviation from baseline EMG amplitude [26,27], or an increase above multiple standard deviations from the average baseline EMG amplitude [28]. In the present study, the onset and offset of the EMG response were determined by a statistical comparison between the EMG amplitude averaged each 5-ms epoch after the onset of platform translation and baseline EMG amplitude (see Figure 1B). This allowed us to determine the presence of the EMG response in the gastrocnemius and soleus muscles with a 5-ms time resolution.

The trigger of the EMG response was determined by the analysis on the EMG amplitude averaged across 40–100, 100–200, or 120–220 ms after the perturbation onset in previous studies [1,2,3]. In the present study, the difference in the EMG amplitude between the ankle conditions was statistically tested at each 5-ms epoch using the multiple comparison test (Ryan’s test) following the test of the simple main effect. This allowed us to determine the precise time window of the EMG response component triggered by the ankle motion and by the body sway above the ankle with a 5-ms time resolution.

### 4.2. Crossed Afferent Inhibition

In the ULL and ULR tasks, the dorsiflexion of the ankle was present in the UM ankle but was absent in the US ankle (see Figure 4B,C). Electrical stimulation of the Ia afferents, via stimulation of the tibial nerve at the popliteal fossa, caused crossed afferent inhibition of the EMG in the contralateral soleus muscle at a latency of 40 ms [29,30]. Thus, crossed afferent inhibition, derived from the moving contralateral ankle (UM ankle), may have inhibited the EMG response in the US ankle. If we consider this as true, the difference in the EMG amplitude between the UM and US ankle could partially be due to crossed afferent inhibition. However, we do not support this view as there was no period showing a significantly lower EMG amplitude after the onset of platform translation relative to the baseline EMG in the US ankle.

### 4.3. Response Preparation

The response to a postural perturbation may be centrally determined [31,32,33], where the central set influences the readiness to respond to a postural perturbation. A previous study reported that the preparatory cortical activity was present before the postural perturbation; the constant warning cue allowed time preparation inducing the contingent negative variation before the postural perturbation, although such a cortical response was absent when the warning cue was not given [31,32]. However, in the present study, a constant warning cue was not presented. Thus, the change in the cortical activity for the time preparation must not have occurred.

The participants knew which side of the support surface under the foot moved before the onset of the platform translation. The advanced knowledge of the moving side of the support surface allowed for event preparation. Thus, one may speculate that the preparation preceding the platform translation may have been different between the tasks. However, the amplitude of the baseline EMG preceding the platform translation was not significantly different between the ankle conditions. Thus, event preparation caused by the advanced knowledge of the moving side did not influence the motor status before the postural perturbation.

### 4.4. Response Triggered by Ankle Motion

The time window, in which the EMG amplitude in the UM ankle was significantly higher than that in the US ankle, began at 75–85 ms after the onset of the platform translation, and ended 125–135 ms after that translation, in the gastrocnemius muscle (see Figure 7). This time window began at 35–50 ms and ended 110–115 ms after the onset of the platform perturbation in the soleus muscle (one exceptional period, 130–135 ms after the onset of platform translation for the right gastrocnemius and soleus muscles for the slow platform translation, was excluded from this discussion). The ankle dorsiflexion was absent in the US ankle but was present in the UM ankle (see Figure 4B,C). Accordingly, the stretch of the tested ankle muscles caused by the ankle motion must have triggered the EMG response component within these time windows.

The short-latency response is induced at a latency of 35–45 ms [7,8,9,10]. Thus, the earliest period of the EMG response component triggered by the ankle motion in the soleus muscle must represent the short-latency response. The onset of the EMG response and the beginning of the time window, in which the EMG amplitude in the UM ankle was significantly greater than that in the US ankle, were earlier for the soleus muscle than for the gastrocnemius muscle (see Figure 7). This supported our hypothesis two, that the latency of the EMG response component triggered by the ankle motion in the soleus muscle is shorter than that in the gastrocnemius muscle, due to different thresholds of the response to the somatic sensation induced by the ankle motion (see Figure 8).

The short-latency response is produced by group Ia muscle afferents [7]. The latency of the T-wave, mediated by the stretch reflex pathway, is about 35 ms [11]. This latency is similar to the latency of the EMG response and the beginning of the time window, in which the EMG amplitude of the soleus muscle for the UM ankle was significantly greater than that of the US ankle. Therefore, the earliest component of the EMG response in the soleus muscle is likely mediated by group Ia afferents contributing to the stretch reflex. The number of muscle spindles in the soleus muscle is greater than that in the gastrocnemius muscle [19], and, consequently, the amplitude of the H-reflex mediated by the group Ia afferents is greater in the soleus muscle than in the gastrocnemius muscle [18]. Accordingly, the difference in the latency of the EMG response, and the difference in the beginning of the time window in which the ankle motion triggers the EMG response, must be derived from different feedback of group Ia afferents between the muscles.

### 4.5. Contribution of Polysynaptic Pathway

The dorsiflexion of the ankle, causing a stretch of the tested muscles, terminated around 60 ms after the onset of the platform translation (see Figure 3). The latency of the T-wave, mediated by the stretch reflex pathway, is about 35 ms [11]. Thus, the EMG response component triggered by the monosynaptic stretch reflex pathway must have occurred until 95–100 ms after the platform translation onset. The time window, in which the EMG amplitude in the UM ankle was significantly greater than that in the US ankle, terminated 110–135 ms after the onset of platform translation (see Figure 7). Thus, the termination of this time window was later than the end of the EMG response component triggered by the monosynaptic stretch reflex. This means that the EMG response component triggered by the ankle motion persisted beyond the completion of the monosynaptic stretch reflex. Thus, hypothesis one, stating that the EMG response component triggered by the ankle motion is not only mediated by the monosynaptic stretch reflex pathway but also by the polysynaptic pathway, is supported (see Figure 8).

A cutaneous sensation is produced during the dorsiflexion of the ankle induced by the translation of the platform due to the deviation of the center of mass causing a moving tactile sensation on the foot sole. The cutaneous sensation changes the magnitude of the response to the postural perturbation in cats [34]. Thus, we cannot rule out a possibility that the EMG response component triggered by the ankle motion was partially mediated by the cutaneous sensation. The pathway mediating the cutaneous pathway is polysynaptic [12]. Accordingly, the latency of the response component triggered by the cutaneous sensation must be longer than the response mediated by the monosynaptic stretch reflex pathway. Such a polysynaptic pathway mediating the cutaneous sensation possibly contributed to the late component of the EMG response triggered by the ankle motion.

### 4.6. Response without Ankle Motion

In previous studies, the EMG response to a postural perturbation, in the time window between 100 and 200 or between 120 and 220 ms after the onset of perturbation, was present even in the nulled ankle input task [1,2,3]. The correction response appeared at a latency of about 110 ms after the onset of the perturbation, even in patients with a loss of lower limb proprioception [3]. These previous findings indicated that the late component EMG response was triggered by the body sway above the ankle. In the present study, the EMG amplitude after the onset of the platform translation was significantly higher than the baseline EMG amplitude in the time window between 95–155 ms after the onset of platform translation in the US ankle, indicating that the EMG response was present in this time window (see Figure 6).

As shown in the velocity of ankle dorsiflexion, ankle motion was absent in the US ankle (see Figure 4B,C). Thus, this late EMG response was present even in the absence of ankle motion, as is consistent with a previous study [6]. Despite the absence of the ankle dorsiflexion, pelvis sway was present in the US ankle (see Figure 4A). Thus, the EMG response component in the time window between 95 and 155 ms after the onset of platform translation was likely triggered by the body sway above the ankle.

### 4.7. Response to Body Sway above Ankle

There were certain time periods in which the EMG amplitude in the B ankles was significantly higher than in the UM ankle, as consistent with a previous finding [6]. The time window mostly began at 95–100 ms and terminated at 145–155 ms after the platform translation onset (except for the 55–60 ms after platform translation onset in the right soleus muscle of the fast condition, which is excluded from this discussion) (see Figure 7). The pelvis sway in the B ankles was significantly greater than that in the UM ankle, although the velocity of the ankle dorsiflexion was similar between the B and UM ankles (see Figure 4). Thus, the time window, in which the EMG amplitude in the B ankles was significantly greater than that in the UM ankle, reflected the period in which the body sway above the ankle triggered the EMG response. This time window was apparently later than the time window in which ankle motion triggered the EMG response. Taken together, ankle motion triggers the early component of the EMG response, but body sway above the ankle triggers the late component of the EMG response.

### 4.8. Middle-Latency Response

Inhibition of the group II afferents, by medication, decreased the amplitude of the middle-latency response [35]. In addition, cooling of the mixed nerve delayed the middle-latency response, indicating that the small diameter group II afferents mediate the middle-latency response [9]. The group II afferents mediate the stretch sensation of the muscle spindles. Thus, the middle-latency response must be involved in the EMG response component triggered by the ankle motion causing ankle muscle stretch.

The latency of the middle latency response was found to be 85–91 ms in the soleus muscle [10] and 88–110 ms in the gastrocnemius muscle [10,26,27,33]. The time window, in which the EMG amplitude in the UM ankle was significantly greater than that in the US ankle, terminated 125–135 ms after the onset of platform translation in the gastrocnemius muscle and 110–115 ms in the soleus muscle (see Figure 7). Accordingly, the end of this time window was later than the onset of the middle-latency response. This indicated that the middle-latency response was, at least partially, at the end of the EMG response component triggered by the ankle motion (see Figure 8).

## 5. Conclusions

The time window of the EMG response component triggered by the ankle motion began at a latency of 75–85 ms after the onset of the platform translation, ending 125–135 ms after the onset of the platform translation for the gastrocnemius muscle and, for the soleus muscle, began 35–50 ms after and ended 110–115 ms after. The beginning of this time window in the soleus muscle was similar to the latency of the short-latency response, indicating that the early EMG response component triggered by the ankle motion represented the short-latency response.

The difference in the beginning of this time window between the gastrocnemius and soleus muscles is likely due to the different threshold of the response to the feedback from the group Ia afferents between the muscles. The EMG response component triggered by ankle motion continued long after the termination of the ankle dorsiflexion, indicating that this response is mediated not only by the monosynaptic stretch reflex pathway but also by the polysynaptic pathway. The EMG response component triggered by the body sway above the ankle began 90–100 ms and ended 145–155 ms after the platform translation onset. Taken together, ankle motion triggered the early component of the EMG response, but the body sway above the ankle triggered the late component of the EMG response.

## Figures and Tables

**Figure 1 brainsci-10-00230-f001:**
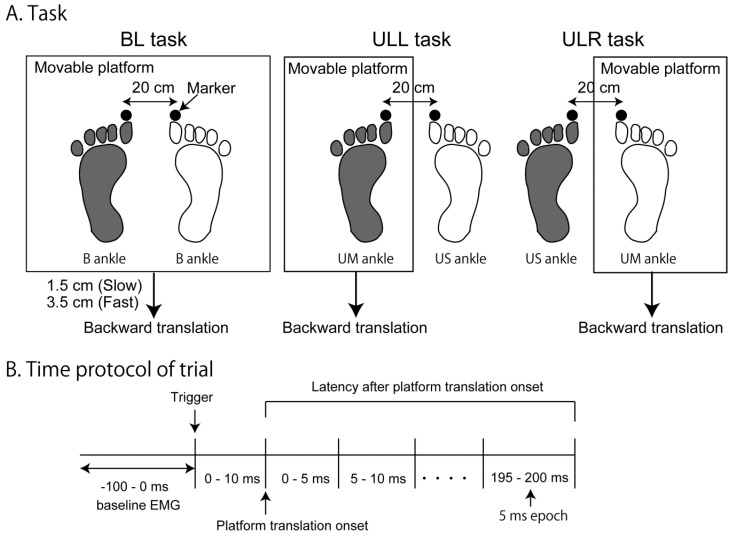
Experimental setup (**A**) and time protocol of the trial (**B**). Each comparison in the electromyographic (EMG) amplitude was conducted in each side of the ankle (one comparison for the filled footprints and another comparison for the open footprints) (**A**). Schematic of the time points for each of the measured variables (**B**). BL, translation of both lower limbs, inducing ankle dorsiflexion bilaterally; ULL, unilateral left ankle dorsiflexion; ULR, unilateral right ankle dorsiflexion; B, bilateral ankle dorsiflexion in BL task; UM, dorsiflexion ankle in either the ULL or ULR task; US, stationary ankle in either the ULL or ULR task.

**Figure 2 brainsci-10-00230-f002:**
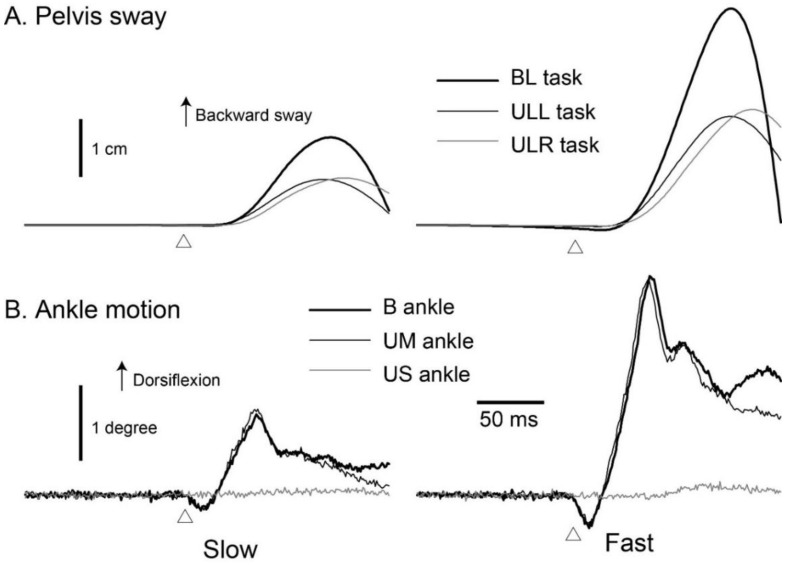
The averaged traces of the pelvis sway (**A**) and ankle motion (**B**) for one participant. Open triangles indicate the onset of platform translation. BL, the translation of both lower limbs, inducing ankle dorsiflexion bilaterally; ULL, unilateral left ankle dorsiflexion; ULR, unilateral right ankle dorsiflexion; B, bilateral ankle dorsiflexion in BL task; UM, dorsiflexion ankle in either the ULL or ULR task; US, stationary ankle in either the ULL or ULR task.

**Figure 3 brainsci-10-00230-f003:**
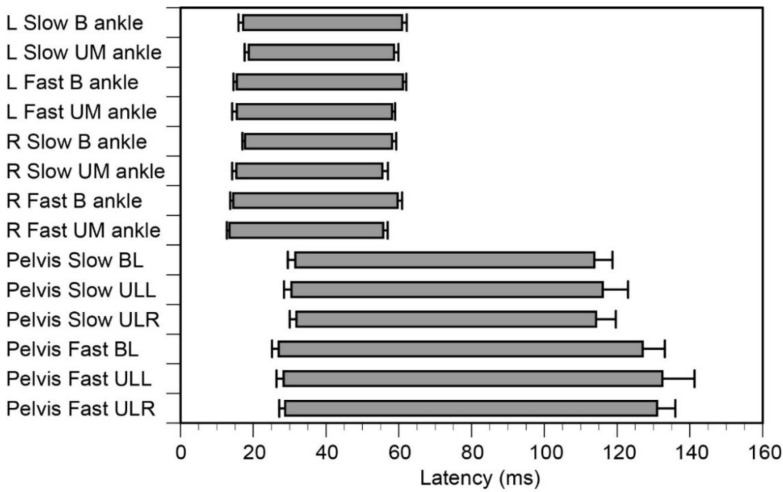
The onset and peak latencies of the ankle dorsiflexion and pelvis sway. Bars indicate the mean, and error bars indicate the standard errors of the mean. The left ends of the bars indicate the latency and the right ends of the bars indicate the peak latency. The top eight bars indicate the ankle motion, and the lower six bars indicate the pelvis sway. L, left; R, right; BL, bilateral; ULL, unilateral left ankle dorsiflexion; ULR; unilateral right ankle dorsiflexion; B, bilateral ankle dorsiflexion in the BL task; UM, dorsiflexion ankle in either the ULL or ULR task.

**Figure 4 brainsci-10-00230-f004:**
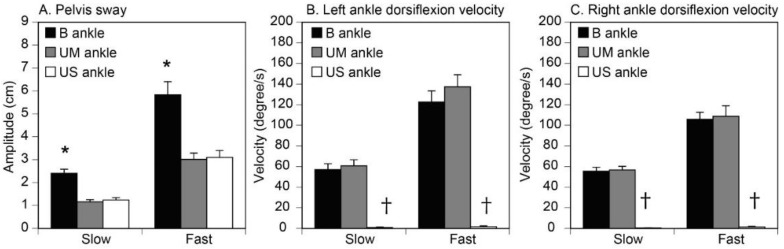
The amplitude of the pelvis sway (**A**), and the velocity of the ankle dorsiflexion (**B**, **C**). Bars indicate the mean, and error bars indicate the standard errors of the mean. (**A**) Asterisks indicate a greater amplitude in the BL compared with the ULL or ULR task on the multiple comparison test (*p* < 0.001). (**B** and **C**) Daggers indicate the lower velocity of dorsiflexion for the US than for the B or UM ankle on the multiple comparison test (*p* < 0.001). B, bilateral ankle dorsiflexion in BL task; UM, dorsiflexion ankle in either the ULL or ULR task; US, stationary ankle in either the ULL or ULR task.

**Figure 5 brainsci-10-00230-f005:**
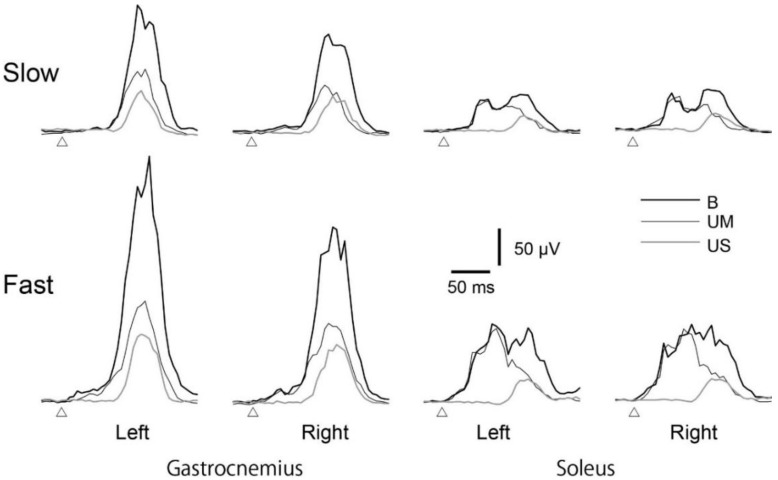
The EMG traces averaged across all participants. Open triangles indicate the onset of the platform translation. B, bilateral ankle dorsiflexion in BL task; UM, dorsiflexion ankle in either the ULL or ULR task; US, stationary ankle in either the ULL or ULR task.

**Figure 6 brainsci-10-00230-f006:**
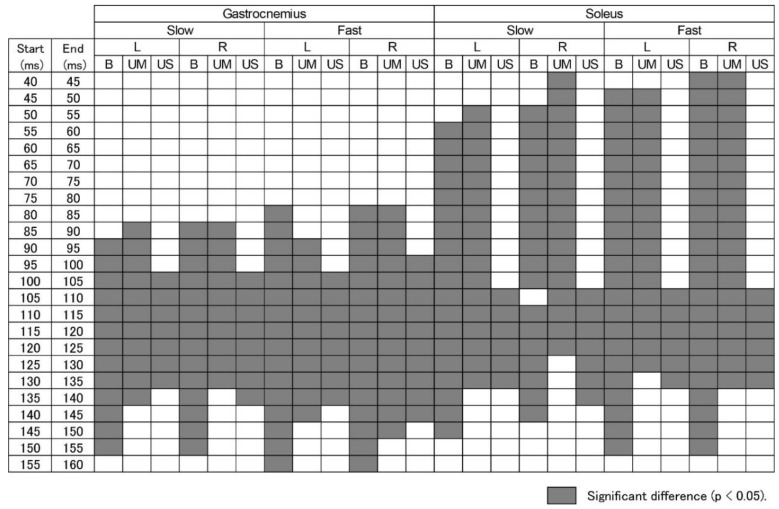
The time windows in which the EMG amplitude was significantly greater than the baseline. The cells before the latency of 40 ms and after the latency of 160 ms are not presented, as there are no time epochs indicating a significant difference in those time windows. Filled cells indicate significant differences (*p* < 0.05). B, bilateral ankle dorsiflexion in BL task; UM, dorsiflexion ankle in either the ULL or ULR task; US, stationary ankle in either the ULL or ULR task; L, left side; R, right side.

**Figure 7 brainsci-10-00230-f007:**
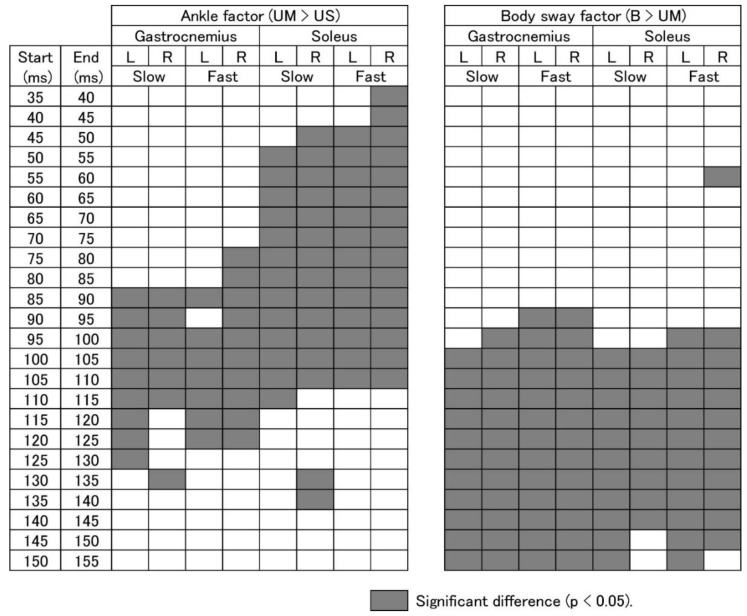
Results of the multiple comparison test for the ankle and body sway factors on the EMG response. The cells before the latency of 35 ms and after the latency of 155 ms are not presented, as there are no time epochs indicating a significant difference in those time windows. The ankle factor is present in the periods in which the EMG amplitude in the UM ankle was significantly higher than that in the US ankle. The body sway factor is present in the periods in which the EMG amplitude in the B ankles was significantly higher than that in the UM ankle. Filled cells indicate significant differences (*p* < 0.05). B, bilateral ankle dorsiflexion in the BL task; UM, dorsiflexion ankle in either the ULL or ULR task; US, stationary ankle in either the ULL or ULR task.

**Figure 8 brainsci-10-00230-f008:**
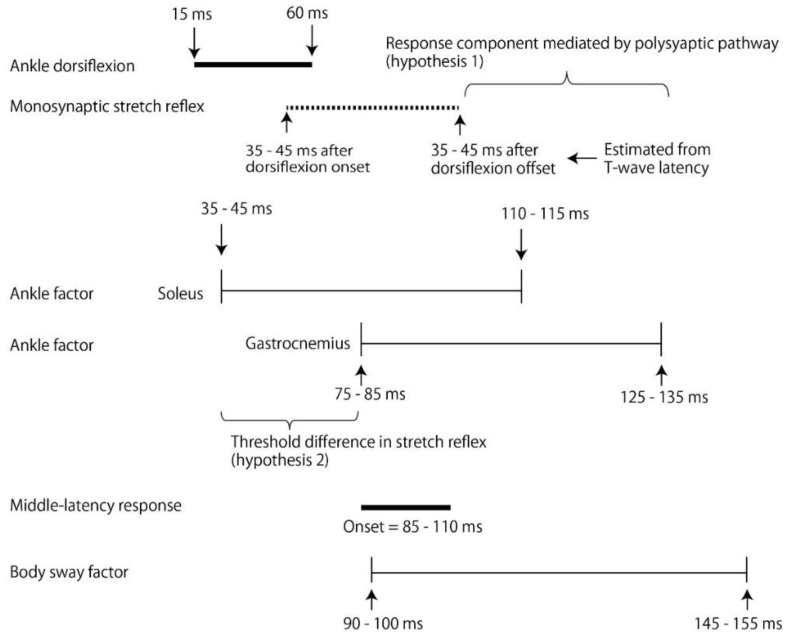
The time events regarding the EMG response. The time periods represent the latency from the platform translation onset. The ankle dorsiflexion was terminated around 60 ms after the platform translation onset, and thus, the stretch of the ankle extensors is terminated at this time. The latency of the monosynaptic stretch reflex is 35–45 ms according to a previous finding (see text), and thus, the response mediated by the monosynaptic reflex must be terminated in the time window between 35–45 ms after the dorsiflexion onset and 35–45 ms after the dorsiflexion offset. The ankle factor continued long after the end of the response mediated by the monosynaptic stretch reflex. According to this finding, the time window of this time factor must be mediated by a polysynaptic response pathway, such as the cutaneous pathway (hypothesis one supported). The beginning of the ankle factor in the gastrocnemius muscle is much later than that in the soleus muscle; however, the beginning is still in the time window of the stretch reflex process. This is explained by a view that the threshold of the response to the somatic sensation of the ankle motion in the gastrocnemius muscle is higher than that in the soleus muscle (hypothesis two supported). The middle-latency response onset with 85–110 ms of the latency (see text). Accordingly, the middle latency response is shown at the end of the ankle factor in the soleus, in the middle of the ankle factor in the gastrocnemius, and at the beginning of the body sway factor.

**Table 1 brainsci-10-00230-t001:** The analysis of variance (ANOVA) on the pelvis sway amplitude.

	df	F	*p*-Value
Task	2	93.379	<0.001
Velocity	1	83.75	<0.001
Interaction	2	42.941	<0.001

**Table 2 brainsci-10-00230-t002:** ANOVAs for the ankle velocity.

		df	F	*p*-Value
Left	Ankle	1	26.571	<0.001
	Velocity	1	128.252	<0.001
	Interaction	1	8.593	0.011
Right	Ankle	1	0.13	0.723
	Velocity	1	84.054	<0.001
	Interaction	1	0.025	0.878

**Table 3 brainsci-10-00230-t003:** ANOVAs on the EMG amplitude.

		Slow			Fast	
		df	F	*p*-Value	F	*p*-Value
Gastrocnemius						
Left	Ankle	2	33.583	<0.001	37.818	<0.001
	Delay	40	25.836	<0.001	24.366	<0.001
	Interaction	80	13.764	<0.001	12.051	<0.001
Right	Ankle	2	34.881	<0.001	47.301	<0.001
	Delay	40	23.561	<0.001	34.565	<0.001
	Interaction	80	12.289	<0.001	15.077	<0.001
Soleus						
Left	Ankle	2	76.22	<0.001	59.532	<0.001
	Delay	40	14.948	<0.001	15.427	<0.001
	Interaction	80	9.616	<0.001	12.891	<0.001
Right	Ankle	2	34.671	<0.001	84.443	<0.001
	Delay	40	24.188	<0.001	20.797	<0.001
	Interaction	80	10.676	<0.001	13.53	<0.001

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
