# Peer review of "Time Window of Perturbation-Induced Response Triggered by Ankle Motion and Body Sway above the Ankle"

_brainsci, 2020, doi:10.3390/brainsci10040230_

Round 1

Reviewer 1 Report

Main points to address

The authors of this study set out to define with greater time resolution muscle activity generated as a result of perturbations and which changes in EMG activity is related to sway produced above the ankle and that produced by movement of the ankle. The authors did a very convincing job at this. The study is clear and nicely written. The reader is nevertheless left wondering what the value of this work really is since the results are quite similar to previous ones in terms of timing of activity (though much less precise approaches were used). Though greatly oversimplified, what did this study achieve with this higher resolution. The authors need to address this point. 

The introduction and discussion should talk about the different roles played by the two muscles from which activity was recorded by the authors. What are their respective functions and why were they chosen for study. In general, the paper lacks explanations in terms of function. What do the results tell us in terms of function? 

Explain also what the different EMG latency across the two muscles means in terms of functions. Why do they have different onsets?     

In terms of methods, please clarify why subjects were asked to close their eyes. Also, please clarify what instructions were given to the participants, e.g. keep your balance, and why perturbations were generated only in one direction and why it was decided to let the subjects know in advance the type of perturbation that would be applied. The reviewer also would like to know whether sway along the mediolateral axis was also measured. It would be nice to report this information as well. 

Please also suggest next steps in the conclusion. Where should we go from here? What other experiments should be conducted to investigate this further? One suggestion for future studies would be to give subjects a secondary task to perform, for example saccades of varying degrees of difficulty and observe how the postural system would behave under dual-task circumstances (see for example Boulanger, Giraudet, Faubert, 2017). Would similar strategies be used?       

Minor corrections Introduction

Lines 34-35: change for “Those response components are mediated by different neural processes, whose latencies vary.”

Line 37: change for “...derived from a different…” + “... it allows to speculate on…”

Line 38: change for “...the body part from which the somatic sensation originates…”

Line 40: change for “...by a somatic…”

Line 41: change for “...from a particular…”

Line 43: change for “...220 ms after it…” + “There are weaknesses in…”

Line 44: change for ”One weakness was…”

Line 45: change for “...triggered by ankle motion…”

Line 123: change for “...muscle either than that…”

Minor corrections Methods

Line 215: change for “...was conducted on…”

Line 216: change for “...for the data on the same side…”

Minor corrections Results

Line 239: change for “...are shown… the average sway onset…”

Line 240: change for “...latency of peak sway across…

Line 245: change for “The latency of peak pelvis sway…”

Line 250-251: change for “The amplitudes were ...in the slow condition, while in the fast condition the amplitudes were...”

Line 298: change for “...and latency of peak ankle…”

Line 310: change for “...dorsiflexion showed a significant…”

Author Response

Summary of revision

We greatly appreciate very important suggestions on our manuscript. We have carefully read the comments and revised our manuscript. In addition, a native English editor edited our revised manuscript. Followings are the summary of our revisions. The revised sentences were highlighted in the text. We would like to revise again if further revisions are needed.

(Comment 1)

The reader is nevertheless left wondering what the value of this work really is since the results are quite similar to previous ones in terms of timing of activity (though much less precise approaches were used). Though greatly oversimplified, what did this study achieve with this higher resolution. The authors need to address this point.

(Response)

Thank you for the important comment. We added a sentence that states the benefit of the estimating the precise time window of the EMG response components (lines 64-67).

(Comment 2)

The introduction and discussion should talk about the different roles played by the two muscles from which activity was recorded by the authors. What are their respective functions and why were they chosen for study.

(Response)

Thank you for the comment. This was additionally mentioned in paragraph 2 in Introduction (lines 33-38).

(Comment 3)

In general, the paper lacks explanations in terms of function. What do the results tell us in terms of function?

(Response)

Thank you for the comment. We discussed the difference in the functional roles for those muscles in Discussion (lines 546-553).

(Comment 4)

Explain also what the different EMG latency across the two muscles means in terms of functions. Why do they have different onsets?

(Response)

Thank you for the comment. We discussed the difference in the functional roles for those muscles in Discussion (lines 546-553).

(Comment 5)

In terms of methods, please clarify why subjects were asked to close their eyes.

(Response)

Thank you for the comment. The participants closed their eyes to exclude the influence of the visual input on the response. We revised the sentence (lines 175-177).

(Comment 6)

Also, please clarify what instructions were given to the participants, e.g. keep your balance, and why perturbations were generated only in one direction and why it was decided to let the subjects know in advance the type of perturbation that would be applied.

(Response)

Thank you. The reason why backward translation was given was additionally mentioned (lines 177-179). Instruction to maintain upright position against the perturbation was additionally mentioned (lines 173-174). The reason for giving information of the direction to the participants was added (lines 171-173).

(Comment 7)

The reviewer also would like to know whether sway along the mediolateral axis was also measured. It would be nice to report this information as well.

(Response)

Thank you. We did not measure the pelvis displacement in the medial-lateral direction. This was due to our view that the backward translation of the platform causes mainly the displacement of the body in the anterior-posterior direction. However, indeed, in the ULL and ULR tasks, the support surface under the foot was translated unilaterally, indicating that the medial-lateral displacement of the body has been possible to be occurred. This was a weakness of the present study, and thus, discussed in Discussion (lines 466-471).

(Comment 8)

Please also suggest next steps in the conclusion. Where should we go from here? What other experiments should be conducted to investigate this further? One suggestion for future studies would be to give subjects a secondary task to perform, for example saccades of varying degrees of difficulty and observe how the postural system would behave under dual-task circumstances (see for example Boulanger, Giraudet, Faubert, 2017). Would similar strategies be used?  

(Response)

Thank you for the comment. We added a paragraph stating the future studies in Conclusion (lines 640-646).

(Comment 9)

Lines 34-35: change for “Those response components are mediated by different neural processes, whose latencies vary.”

(Response)

Thank you for the comment. We revised the sentence as the reviewer suggested (lines 43-44).

(Comment 10)

Line 37: change for “...derived from a different…” + “... it allows to speculate on…”

(Response)

Thank you for the comment. We revised the sentence as the reviewer suggested, and a native English editor revised later (line 46).

(Comment 11)

Line 38: change for “...the body part from which the somatic sensation originates…”

(Response)

Thank you for the comment. We revised the sentence as the reviewer suggested (lines 46-47).

(Comment 12)

Line 40: change for “...by a somatic…”

(Response)

Thank you for the comment. We revised the sentence as the reviewer suggested (line 48).

(Comment 13)

Line 41: change for “...from a particular…”

(Response)

Thank you for the comment. We revised the sentence as the reviewer suggested (lines 45, 49).

(Comment 14)

Line 43: change for “...220 ms after it…” + “There are weaknesses in…”

(Response)

Thank you for the comment. We revised the sentence as the reviewer suggested, and a native English editor slightly changed it (lines 50-51).

(Comment 15)

Line 44: change for ”One weakness was…”

(Response)

Thank you for the comment. We revised the sentence as the reviewer suggested (line 52).

(Comment 16)

Line 45: change for “...triggered by ankle motion…”

(Response)

Thank you for the comment. We revised the sentence as the reviewer suggested (lines 52-53).

(Comment 17)

Line 123: change for “...muscle either than that…”

(Response)

Thank you for the comment. We revised the sentence as the reviewer suggested (line 134).

(Comment 18)

Line 215: change for “...was conducted on…”

(Response)

Thank you for the comment. We revised the sentence as the reviewer suggested (lines 233, 234).

(Comment 19)

Line 216: change for “...for the data on the same side…”

(Response)

Thank you for the comment. We revised the sentence as the reviewer suggested (line 234-235).

(Comment 20)

Line 239: change for “...are shown… the average sway onset…”

(Response)

Thank you for the comment. We revised the sentence as the reviewer suggested (lines 258-259).

(Comment 21)

Line 240: change for “...latency of peak sway across…

(Response)

Thank you for the comment. We revised the sentence as the reviewer suggested (line 259).

(Comment 22)

Line 245: change for “The latency of peak pelvis sway…”

(Response)

Thank you for the comment. We revised the sentence (line 265).

(Comment 23)

Line 250-251: change for “The amplitudes were ...in the slow condition, while in the fast condition the amplitudes were...”

(Response)

Thank you for the comment. We revised the sentence as the reviewer suggested, and a native English editor revised it (lines 269-271).

(Comment 24)

Line 298: change for “...and latency of peak ankle…”

(Response)

Thank you for the comment. We did not change it, because this sentence means two different latencies; onset latency and peak latency (line 312).

(Comment 25)

Line 310: change for “...dorsiflexion showed a significant…”

(Response)

Thank you for the comment. We did not change this to keep our intended mean on this sentence (lines 327-330).

Reviewer 2 Report

On the whole we considered this paper well considered and nicely executed. The experimental set-up is ingenious in separating contributions of ankle and pelvic movement. Our main comments are two.

  1. Line 54. The many abbreviations are confusing. Restrict them to only long expression that occur very frequently. I suggest to leave only the experimental tasks: B, ULL, ULR. In the figures you can use a few more, when they are explained in the legends.
  2. The results of the ANOVA are reported in full everywhere. It is advised to first report the findings in short, and only later to give the statistics, and only where needed. For example, in paragraph 249-259. “…….The amplitude was 2.4 ± 0.2 cm in the bilateral task and half this value, 1.2 ± 0.1 cm for the unilateral tasks, both left and right, for the slow translation. ………” That the bilateral amplitudes are higher than the unilateral ones, is the express purpose of the experimental set-up, and does not need extensive statistics.
    Detailed comments.
  3. Line 28. This sentence is not correct, the center of pressure usually does not leave the base of support (= the area between both feet). “During quiet standing the center of pressure moves in a small area in the center of the base of support. With a postural perturbation………..
  4. L 147 Specify the positions of the EMG electrodes more specifically. Was the medial or lateral part of soleus used? Were the electrodes placed in a rostro-caudal or medio-lateral direction?
  5. 243 ‘The onset latencies for pelvic sway were different for different velocities.’ How much? I cannot see it from figure 2A. According to Fig 3 the difference is minimal.
  6. 249-253 That the amplitudes of both ankle and pelvic sway depend on velocity, will have to do with the different displacements, 15 and 35 mm, respectively (line 166-167).
  7. 356 Best to start with the major conclusion: onset time for soleus was 40-50 ms, for gastrocnemius 80 ms. Then differences UM/US. Then other details. Only then statistics.

Author Response

Summary of revision

We greatly appreciate very important suggestions on our manuscript. We have carefully read the comments and revised our manuscript. In addition, a native English editor edited our revised manuscript. Followings are the summary of our revisions. The revised sentences were highlighted in the text. We would like to revise again if further revisions are needed.

(Comment 26)

Line 54. The many abbreviations are confusing. Restrict them to only long expression that occur very frequently. I suggest to leave only the experimental tasks: B, ULL, ULR. In the figures you can use a few more, when they are explained in the legends.

(Response)

Thank you for the practical comment. We used “platform” instead of “PF”, and used “unilateral tasks” instead of “UL tasks” in the present version.

(Comment 27)

The results of the ANOVA are reported in full everywhere. It is advised to first report the findings in short, and only later to give the statistics, and only where needed. For example, in paragraph 249-259. “…….The amplitude was 2.4 ± 0.2 cm in the bilateral task and half this value, 1.2 ± 0.1 cm for the unilateral tasks, both left and right, for the slow translation. ………” That the bilateral amplitudes are higher than the unilateral ones, is the express purpose of the experimental set-up, and does not need extensive statistics.

(Response)

Thank you for the comment. As the reviewer suggested, we mentioned the mean values first, and then, statistics were mentioned next in Results in the revised version (lines 262, 265-266, 315-316, 319, 337-338, 340-341).. For the pelvis sway amplitude, we had to conduct the statistical analysis to confirm whether the pelvis sway in the B was significantly greater than the UM or US, and that was not significantly different between the UM and US.

(Comment 28)

Line 28. This sentence is not correct, the center of pressure usually does not leave the base of support (= the area between both feet). “During quiet standing the center of pressure moves in a small area in the center of the base of support. With a postural perturbation………..

(Response)

Thank you for the comment. We changed the phrase as “center of pressure goes to the border of the base of support” (line 28).

(Comment 29)

L 147 Specify the positions of the EMG electrodes more specifically. Was the medial or lateral part of soleus used? Were the electrodes placed in a rostro-caudal or medio-lateral direction?

(Response)

Thank you for the comment. The EMG was recorded from the medial gastrocnemius muscle (lines 159-161). The electrodes measuring the EMG signals from the soleus muscle was placed slightly medial to the medial border of the gastrocnemius muscle (lines 161-163). Those sentences were revised or added.

(Comment 30)

243 ‘The onset latencies for pelvic sway were different for different velocities.’ How much? I cannot see it from figure 2A. According to Fig 3 the difference is minimal.

(Response)

Thank you for the comment. The difference was as small as 3 ms. The mean values were additionally mentioned before the statistical results in Results (lines 262, 265-266, 315-316, 319, 337-338, 340-341).

(Comment 31)

249-253 That the amplitudes of both ankle and pelvic sway depend on velocity, will have to do with the different displacements, 15 and 35 mm, respectively (line 166-167).

(Response)

Thank you for the comment. To match the duration of the translation, the amplitude was changed. The amplitude of the pelvis sway and the velocity of the ankle dorsiflexion were greater when the velocity of the platform translation was greater. In spite of that, the amplitude of the PF translation was smaller for the fast condition. That is, the amplitude of the pelvis sway and the velocity of the ankle dorsiflexion was greater when the amplitude of the platform translation was smaller. Thus, we do not think that the amplitude of the translation has to do with the difference in the amplitude of the pelvis sway and the velocity of the ankle motion.

(Comment 32)

356 Best to start with the major conclusion: onset time for soleus was 40-50 ms, for gastrocnemius 80 ms. Then differences UM/US. Then other details. Only then statistics.

(Response)

Thank you for the comment. It is difficult to place the conclusion first in this paragraph, because those values, 40-50 ms and 80 ms, were derived from the results of the multiple comparison following the test of simple main effect after finding the significant interaction between the main effects by ANOVA.
